

# Pseudo-random modulation continuous wave narrowband sodium temperature/wind lidar

Xin Fang[1,2,3#], Feng Li[1#], and Lei-lei Sun[1], Tao Li[1,2,3, *]

[1]Deep Space Exploration Laboratory/School of Earth and Space Sciences, University of Science and Technology of China, Hefei, Anhui, China

[2]CAS Key Laboratory of Geospace Environment, School of Earth and Space Sciences, University of Science and Technology of China, Hefei, Anhui, China

[3]CAS Center for Excellence in Comparative Planetology, University of Science and Technology of China, Hefei, Anhui, China

*# These authors contributed equally to this work*

*Correspondence to: litao@ustc.edu.cn*

**Abstract:** We report the first Pseudo-random Modulation Continuous Wave (PMCW) narrowband sodium temperature/wind lidar developed at the University of Science and Technology of China (USTC). The laser system uses an 1178nm diode seed-laser and a fiber Raman amplifier with a fiber-coupled Acoustic Optical Modulator (AOM) to generate a narrowband 589.158 nm light with a power output of 1.5 W at $v_0$, $v_+$ and $v_-$ frequencies. Based on an innovative technique and algorithm, the main beam and the residual beam modulated by Electro-Optic Modulator (EOM) with M-code are separately directed to the vertical and eastward directions. The 3-frequency light is designed in timing with the multiple-period 127-bit M-code groups. The uncertainties of the temperature and wind with the vertical and temporal resolutions of 1 km and 30 min/1hr under the clear-sky condition are estimated to be 5.0 K and 10 m/s, respectively at the sodium peak. The temperature and wind results are in good agreement with those observed by satellite and nearby ground-based meteor radar, demonstrating the reliability of the PMCW narrowband sodium lidar system for measuring mesopause region temperature and wind.

## 1. Introduction

Temperature and wind are important atmospheric parameters for studying wave dynamics in the mesopause region, such as gravity waves, atmospheric tides, and planetary waves. Pulsed narrowband sodium lidar can measure the mesopause region temperature and wind with high temporal and vertical resolutions. With the development of laser technology, pulsed narrowband sodium lidar has been




gradually developed and improved. Gibson and Thomas first measured sodium ground-state hyperfine

structure and the mesopause temperature with an accuracy of 15K near the peak of the sodium layer

(Gibson et al., 1979). Using a dye laser pumped by excimer, Fricke and von Zahn (1985) obtained 5K

accuracy of temperature measurements with a vertical resolution of 1km and a temporal resolution of 10

minutes .

She et al. developed a high spectral resolution narrowband sodium lidar system using dual-frequency

technology (She et al., 1992). Subsequently, 3-frequency technology was applied in their lidar for

simultaneous temperature and wind measurements in the MLT region (She et al., 2003).. Hu et al.

reported a mobile narrowband sodium lidar with temperature measurements (Hu et al., 2011). Li et al.

constructed a narrowband sodium lidar with several technical improvements and obtained simultaneous

temperature and wind results (Li et al., 2012). Flashlamp pumped solid-state 589nm laser or solid-state

589nm seed laser based on Raman fiber amplifier was applied in pulsed narrowband sodium lidar (Xia

et al., 2017; Yang et al., 2018). All-solid-state sodium lidar system based on the sum frequency of pulsed

1064nm and pulsed 1319nm were developed to detect the mesopause region temperature and wind

(Kawahara et al, 2002, 2017).

However, pulsed lasers, either dye or all solid in the pulsed sodium lidars, cannot meet the space-borne

platform with size, weight, and power consumption limitations. Therefore, developing a lightweight and

small-size narrowband sodium lidar is necessary for airborne and space-borne purposes. Referring to

the idea of continuous wave (CW) in the application of microwave radar (Ridenour et al., 1948), CW

lidar is a feasible miniaturized and lightweight lidar system. Takeuchi et al. (1986) first developed a CW

aerosol lidar with ranging capability using M pseudo-random sequence code (M-code) modulated CW

laser sources. Abo and Nagasawa (1984) proposed a random modulation CW lidar for Na layer detection .

She et al. (2012) gave a detailed simulation about pseudorandom modulation continuous-wave (PMCW)

sodium lidar for temperature and wind measurements in the mesopause region. The authors pointed out

that PMCW sodium lidar can achieve a signal-to-noise ratio comparable to pulsed sodium lidar. Li et al.

(2021) implemented a PMCW single-frequency sodium lidar for sodium density measurement using an

M-code modulated diode 589nm laser in 2021.

This study aims to develop a PMCW 3-frequency narrowband sodium lidar for simultaneous temperature

and wind measurements in the mesopause region. The detection principle of PMCW narrowband sodium

lidar (PMCW-NSL) is described in section 2. The system structure of PMCW-NSL is presented in section



3. Section 4 discusses the returning signals and temperature and wind results, followed by a summary in section 5.

## 2. The detection principle of PMCW-NSL

The PMCW-NSL mainly utilizes pseudorandom M-code $a_i$ to modulate the CW laser beam and decodes returned signals by the cross-correlation between $a_i$ and its inverse code $a_i'$. The cross-correlation of the M-code has been described in the early simulation by She et al. (2012) and sodium density measurements by Li et al. (2021). It can be given out as below,

$$\phi_{aa'}(k) = \sum_{i=0}^{N-1} a_i a_{i+k}' = \begin{cases} \dfrac{(N+1)}{2} & (k=0) \\ 0 & (k \neq 0) \end{cases} \tag{1}$$

According to the cross-correlation, we can obtain the decoded (inverted) signal with range resolution information as follows,

$$R_j = \frac{(N+1)}{2} \frac{P_0}{h\nu} G_j + B \tag{2}$$

where $P_0$ is the CW laser power; $h\nu$ is the energy of a single photon; $G_j$ (Li et al., 2021) is the atmospheric response of the lidar system at altitude $h_j$ ($h_j = j \cdot c \cdot \Delta t/2$, $c$ is the speed of light and $\Delta t$ is the length of time for a single code $\alpha_i$ ) and $B$ is the average intensity of background noise in $\Delta t$.

When the M code value is 1, the CW beam is transmitted from the polarization crystal inside the electro-optic modulator (EOM). When the M code value is 0, the CW laser is reflected by the polarization crystal. We call the transmitted output light the main light and the reflected output light the residual light. Since the M-code has approximately the same number of occurrences of 0 and 1 in one cycle, this means that half of the energy of the CW laser is lost due to modulation. To improve the utilization efficiency of laser power, an innovative method is proposed for residual light detection under certain conditions and realizing the complete use of CW laser.

Since the principle of CW modulation by EOM is based on the assumption that there is no light loss under ideal conditions, the sum of the main and residual light should equal the intensity of the incident light (West et al., 1951). If the modulation sequence of the main light is the M-code sequence $a_i$, and the residual light sequence is $b_i$, then the relationship between $b_i$ and $a_i$ is given below:

$$b_i = 1 - a_i \tag{3}$$

Similar to the M-Code sequence, we also establish the inverted code of the residual light sequence as:

$$b_i' = 2b_i - 1 \tag{4}$$



By substituting Equation (3) into (4), we can obtain

$$b_i' = 2(1 - a_i) - 1 = -a_i' \tag{5}$$

where $a_i'$ is the inverted code for $a_i$. The cross-correlation function between the residual light sequence

and its inverted code can be calculated according to Equations (3) and (5):

$$\emptyset_{bb'}(k) = -1 + \sum_{i=0}^{N-1} a_{i+k} \, a_{i'} \tag{6}$$

By substituting the cross-correlation function (Equation (1)) of the M sequence into the above equation,

the cross-correlation function of the residual light code sequence can be obtained:

$$\phi_{bb'}(k) = \begin{cases} \dfrac{(N-1)}{2} & (k = 0) \\ -1 & (k \neq 0) \end{cases} \tag{7}$$

Table 1 shows the values of the residual light code, inverted code, and cross-correlation function. The

inverted codes satisfy the autocorrelation property of pseudo-random coding. Although the correlation

function results differ from the M-code, they can also be used for CW laser detection under certain

conditions.

Table 1 Sequences of residual light code, its associated receiving code, and their correlation for N = 31

| $i$ | 0 | 1 | 2 | 3 | 4 | 5 | 6 | 7 | 8 | 9 | 10 | 11 | 12 | 13 | 14 | 15 |
|---|---|---|---|---|---|---|---|---|---|---|---|---|---|---|---|---|
| $b_i$ | 0 | 0 | 0 | 0 | 0 | 0 | 0 | 1 | 0 | 1 | 0 | 1 | 0 | 1 | 1 | 0 |
| $b_i'$ | -1 | -1 | -1 | -1 | -1 | -1 | -1 | 1 | -1 | 1 | -1 | 1 | -1 | 1 | 1 | -1 |
| $\phi_{bb'}(i)$ | 63 | -1 | -1 | -1 | -1 | -1 | -1 | -1 | -1 | -1 | -1 | -1 | -1 | -1 | -1 | -1 |
| $i$ | 16 | 17 | 18 | 19 | 20 | 21 | 22 | 23 | 24 | 25 | 26 | 27 | 28 | 29 | 30 | 31 |
| $b_i$ | 0 | 1 | 1 | 0 | 0 | 0 | 1 | 0 | 0 | 0 | 1 | 0 | 1 | 1 | 0 | 1 |
| $b_i'$ | -1 | 1 | 1 | -1 | -1 | -1 | 1 | -1 | -1 | -1 | 1 | -1 | 1 | 1 | -1 | 1 |
| $\phi_{bb'}(i)$ | -1 | -1 | -1 | -1 | -1 | -1 | -1 | -1 | -1 | -1 | -1 | -1 | -1 | -1 | -1 | -1 |

Similar to the M sequence, residual light is modulated by the residual light sequence $b_i$ to CW, and then

the acquisition signal equation can be obtained:

$$S_i = \frac{P_0}{h\nu} \sum_{k=0}^{N-1} (G_k b_{i-k}) + B \tag{8}$$

For a single acquisition cycle time, the sky background noise signal is nearly constant, and the system

noise signal is replaced with a constant signal. Similar to the main light, the acquired signal above is

decoded by the one-to-one inverse code of the residual light sequence, and the new inverted signal $R_m$

at altitude $h_m$ ($h_m = m \cdot c \cdot \Delta t / 2$, where $c$ is the speed of light and $\Delta t$ is the length of time for a single



code $b_{i'}$) can be obtained as follows:

$$R_m = \frac{P_0}{hv} \sum_{i=0}^{N-1} (b'_{i-m} S_i) = \frac{P_0}{hv} \frac{(N+1)}{2} G_m - \frac{P_0}{hv} \sum_{k=0}^{N-1} G_k - B \qquad (9)$$

Comparing the decoded signals from the M code, we can find several differences in the final form of Equation (9) from Equation (1): there is an additional term in the decoded signal. This term is always a negative constant when computing the decoded signal of any given height. The background signal is also negative. The decoded signals are just the altitude-returned signals superimposed with a large negative background. Consequently, when CW lidar detection is performed using residual light, the background noise cannot be directly acquired in the decoded signal. To solve this issue, we add an additional zero-set code period to the M code to explicitly measure the background noise signal.

## 3. PMCW-NSL system

The PMCW-NSL system consists of a transmitter with a CW laser modulated by an M-code, a receiver with two Newtonian telescopes, and a specifically designed acquisition and timing subsystem. The schematic diagram of the PMCW-NSL system is shown in Figure 1.

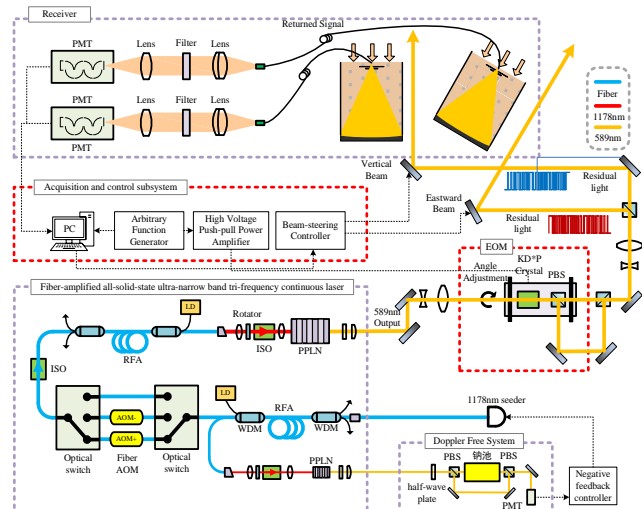

Figure 1 Schematic diagram of PMCW -NSL system

The transmitter is to generate an M-code modulated 589nm CW laser and send it into the atmosphere. In cooperation with Shanghai Frequency Calibration Co., LTD., we have developed an all-solid-state CW



laser source suitable for PMCW-NSL based on fiber Raman laser technology (Murray et al., 1998; Feng et al., 2004; Taylor et al., 2010)). Its basic principle is shown in Figure 1, and its physical photo is shown in Figure 2. A narrowband diode 1178nm continuous laser with a linewidth of ~50kHz and power of 30 mW (TOPTICA DL Pro, tunable diode laser) acts as the seeder injected into the fiber Raman amplifier with two-stage fiber Raman amplification. The first stage of the fiber Raman amplifier amplifies the 1178 nm seed laser to about 150 mW, divided into two outputs, one for laser frequency locking and the other for the input of the secondary amplification after frequency shift by fiber Acoustic-Optic-Modulation (AOM). The laser frequency locking is completed by a Doppler-free saturated absorption system with an accuracy of $\pm 1$MHz. The fiber AOM consisted of two AOMs and two optical switches shift laser frequency 0 MHz, +315 MHz, and -315 MHz respectively at 1178.316 nm (corresponding frequency shifts 0 MHz($v_0$), +630 MHz($v_+$) and -630 MHz($v_-$) at 589.158nm after passed the Second-Harmonic-Generator (SHG) called Periodically polarized lithium niobate (PPLN)). The power of an amplified 589 nm laser is ~1.5 W at $v_0$, $v_+$ and $v_-$ frequencies. Notably, the optical fiber Raman amplifier only weighs ~25kg and consumes power ~450 Watts, clearly superior to the laser used in pulsed sodium lidar.

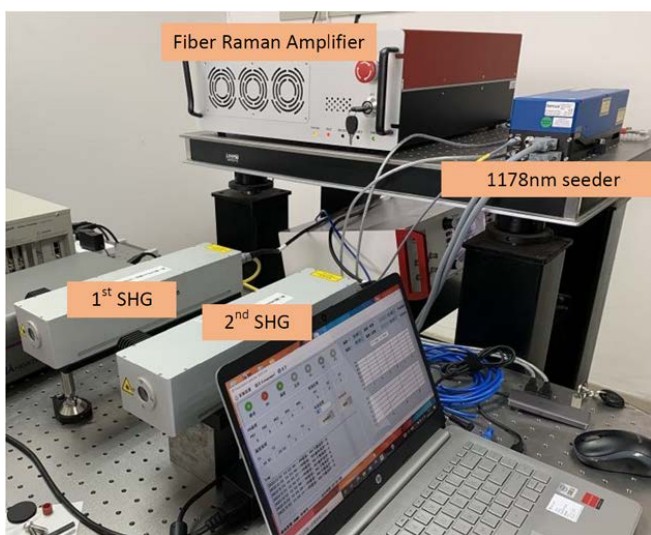

Figure2 Photo of the fiber Raman amplifier

M-code then modulates the amplified 589nm laser via an electro-optic modulator (EOM). Both the main light and the residual light are sent into the atmosphere for respective detection in the vertical and eastward directions. To eliminate the effects on inverted signals of the strong Rayleigh returned signal



from the lower atmosphere called "ghost targets," we deploy the emitting laser at a distance of 9m from

the receiving telescope and align the sending-receiving optics axis by our automatic collimation system

with an accuracy of ~10 $urad$ [6]. The key parameters of the PMCW-NSL system are given in Table 2.

**Table 2 Technical specifications of the PMCW -NSL**

| Transmitter | | |
|---|---|---|
| Seeder | Center wavelength | 1178.316nm |
| | Coarse tuning range | 1178nm±4nm |
| | Mode hop-free tuning range | 10GHz |
| | Power | 50mW |
| | linewidth | 50kHz |
| | Output interface | Fiber coupler |
| Fiber Raman amplifier include fiber AOM | Center wavelength (nm) | 1178.316nm |
| | Operation mode | Continuous wave |
| | AOM frequency modulation mode | Fixed frequency |
| | AOM frequency modulation value | 0, +315MHz, -315MHz @1178nm ( Corresponding to 0, +630MHz, -630MHz@589nm) |
| | AOM frequency switching rising and falling time | <1us |
| | First 1178 nm laser power (W) | 7.4 |
| | First 589 nm laser power (W) | 1.5, 1.5,1.5 @0, +630MHz, -630MHz |
| | Second 589 nm laser power (mW) | 50 |
| | Power adjustment range (%) | 10-100 |
| | Power stability of the first 589 nm laser (%) | RMS: 0.68 |
| | Cooling | Air |
| | Electrical power | 220V, 2A |
| | Weight | 25 kg |
| EOM | Crystal type | KD*P Crystal |
| | Aperture | <2.7mm |
| | Coating | 400-800 nm |
| | Transmission | >85% |



| | Extinction ratio | 300:1-500:1 |
|---|---|---|
| | Bandwidth | DC-to-30MHz |
| | Rise Time (10%-90%) | 8ns |
| **Receiver and acquisition subsystem** | | |
| Telescope | Type | Newtonian primary focus |
| | Diameter | 30" |
| | F/# | F/2 for vertical and F/2.4 for Eastward |
| Fiber | Length | 15m |
| | Core diameter | 1.5mm |
| | Numerical Aperture | 0.37 |
| Interference filter | Central wavelength | 589.1nm |
| | Bandpass | 0.2nm |
| | Transmission | ~70% |
| Detector | Model | H7421 |
| | Peak sensitivity wavelength | 580nm |
| | Quantum efficiency | ~40%@580nm |
| Photon counter | Model | P7882 |
| | Typical counting rate | 350MHz |

The receiver of PMCW-NSL includes two Newtonian prime-focus telescopes to receive the backscattered
returned signals. One is set in the zenith direction to receive the returned signal of the main light; the
other is oriented eastward at 20° tilted from the zenith to receive the returned signal of the residual light.
The returned signal from each telescope is acquired after 10m length optic fiber transmission, lens
collimation, filtering by an optical filter, lens convergence, and photoelectric conversion process. The
main parameters of the receiver are shown in Table 1.

The timing control for the PMCW-NSL is one of the major technical challenges ensuring lidar obtains
returned signals effectively. It is mainly composed of two parts. One is responsible for coding the CW
laser and trigging the acquisition; the other is for switching the laser beam, and tagging returned signals
in 3 frequencies. The timing diagram of PMCW -NSL is shown in Figure 3. A TTL signal(S1) with a
frequency of 3Hz generated by the signal generator acts as the main synchronization clock of the system.
S1 triggers one self-developed FPGA circuit board at a frequency division of 3 to generate 3 TTL signals
(S2, S3, and S4) with a pulse width of S1 period length as the control signals of 3-frequency switching
of fiber AOM and the spectrum tagging signals of photon counter. S1 also externally triggers a DG645



to generate 23 TTL pulsed signals as the external trigger of the photon counter and an arbitrary waveform

generator (AWG). The AWG produces one M-code group with 14 periods 127 M-code and 2 periods

zero-set 127 M-code and a short idle time in one cycle. A 7 $\mu s$ time length for a single code corresponds

to a range resolution of 1.05 km.

The total detection range of the 127 M code is, therefore, 127×1.05≈133 km. The time length of 23 M-

code groups is just slightly less than the S1 period. PMCW-NSL alternatively sends a laser in M-code

groups of 3-frequency in 3 Hz and receives the respective returned signal. The detailed design of the M-

code group was illustrated in our previous work [15]. It is noted that the S5 and S6 signals in one S1

cycle are zoomed in in Figure 3.

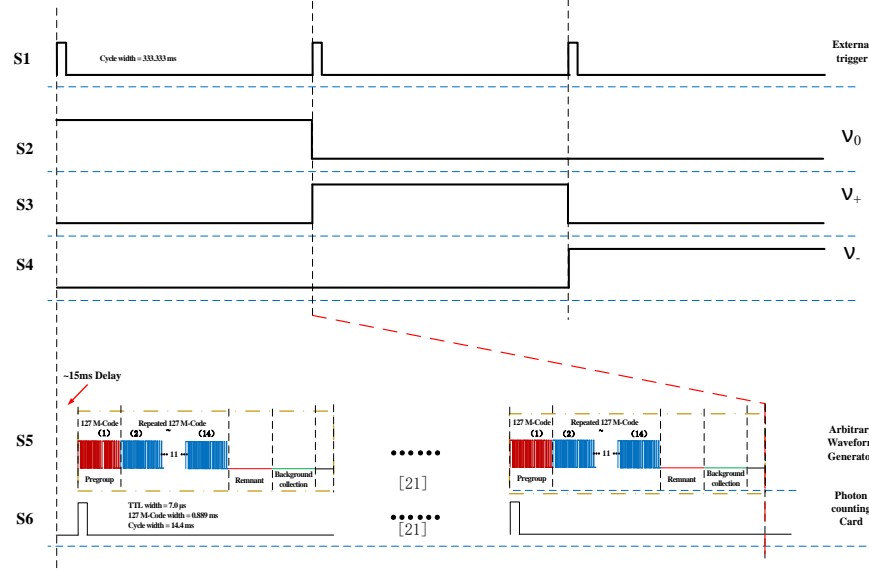

Figure 3 Timing diagram of PMCW -NSL

**4.   Signals and results**

PMCW-NSL acquires backscattering return signals of 3 frequencies ($v_0$, $v_+$ and $v_-$) in five minutes with

2048 bins and 7us bin width. We first sum up signals of 127 M-code cycles at the same code bin in five

minutes and then decode these signals to obtain range-resolved profiles. An example of the 3-frequency

signals observed by PMCW -NSL that accumulated from 10:22UT to 10:26UT on November 24, 2021,

is shown in Figure 4. Figure 4(a) shows the raw 3-frequency signals with a spatial resolution of 1.05km





and a temporal resolution of 5 min. According to equation (7), the inverted signal profile with range resolution information and background removal similar to that obtained by pulsed sodium lidar is shown in Figure 4(b). It is noted that the removal background includes background shift and pure background shot noise acquired at the second period of zero-set 127 M codes. Figure 4(c) and Figure 4(d) are the signal profiles in the sodium layer and corresponding signal-to-noise ratios (SNRs). The number of returned signal photons in 3 frequencies reach $1\times10^5$, $5\times10^4$, and $5\times10^4$, respectively, at the peak altitude of the sodium layer with SNRs larger than 30, 15, and 15 respectively.

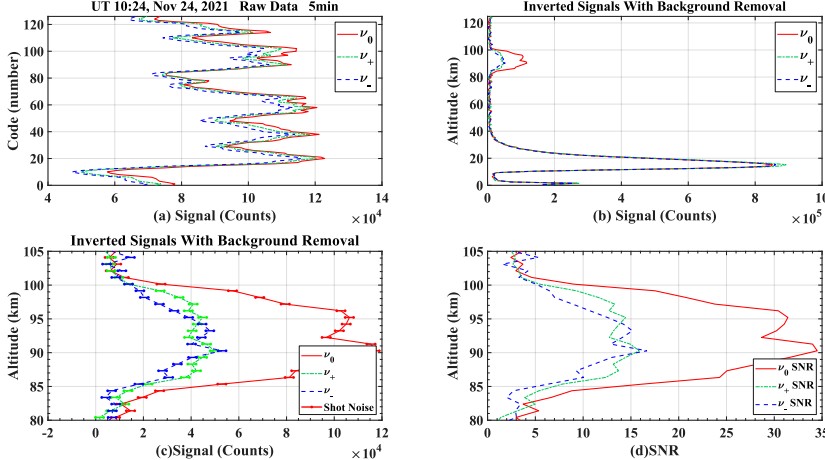

Figure 4 Profiles of (a) raw signals arranged in m-code sequence with 1.05 km spatial and 5 min temporal resolutions on November 24, 2021, (b) inverted signals with background subtracted, (c) inverted signals of only the sodium layer region, and (d) the signal-to-noise ratio of at $\nu_0$, $\nu_+$, $\nu_-$ frequencies of PMCW -NSL.

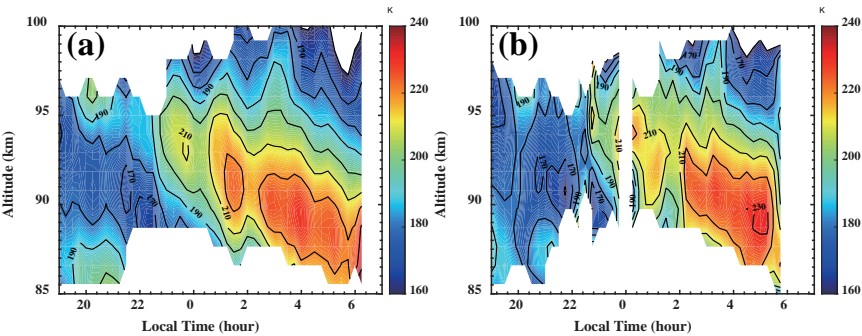

Figure 5 Temperature contour observed by PMCW-NSL on November 24, 2021 in (a)vertical and (b)eastward directions.




Since the SNR of PMCW-NSL is lower than that of pulse lidar at the same power, photon counts are accumulated for at least 15 minutes to improve the SNR. Figure 5 shows the nighttime temperature contour results with a time resolution of 15 min. The temperature varied from 160 to 240K during the night, and an apparent wave structure with a downward phase can be seen, suggesting possible

modulation by tides. The white part in the contour indicates that the temperature uncertainty is greater than 25 K or the signal SNR is insufficient, and the temperature result is invalid. The temperature uncertainty is about 5 K near the peak of the sodium layer (90-95 km), while larger than 25 K at the edge of the sodium layer.

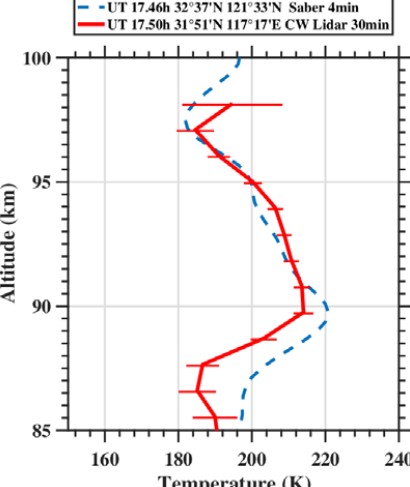

Figure 6 Comparison of temperature profiles between PMCW -NSL (red solid line) and SABER (blue dashed line).


Because our pulsed narrowband sodium temperature/wind lidar cannot simultaneously run with PMCW-NSL, we compare the PMCW-NSL temperature results at 12:30 UT (20:30 LT) with the SABER

observation on November 30, 2021, as shown in Figure6. The temporal and vertical resolutions of PMCW-NSL results are 30 minutes and 1km, respectively, while those for SABER results are 1 minute and 2 km. The temperature observed by the PMCW-NSL is in generally good agreement with the SABER. But there are slight differences in absolute values, significantly below 90km and above 97km. One possible reason for these differences could be the difference in measurement locations and resolutions.



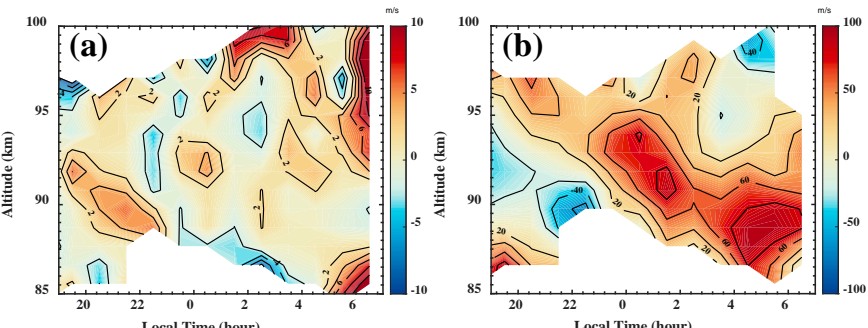


Figure 7 (a)Vertical wind and (b)zonal wind observed by the PMCW-NSL on November 24, 2021.

The vertical and zonal winds observed by PMCW-NSL on November 24, 2021, are shown in Figure 7.
The vertical resolution of the data is 1 km, and the temporal resolution is 1 hour. The white region in the

contour indicates that the wind error at this point is greater than 20m/s, which is considered an unreliable
result. The Figure7(b) shows that the zonal winds at 87-97 km are about -20~100 m/s, and the error is
large at the edge of the sodium layer due to low SNR. From the zonal wind, the contour shows an obvious
downward phase progression, likely related to the solar tides. From the vertical wind profiles shown in
Figure 8, the vertical wind fluctuates within ±5 m/s overnight, and the nightly mean of the vertical wind

velocity is close to 0. This is in full agreement with other vertical wind observations (White et al., 1999).

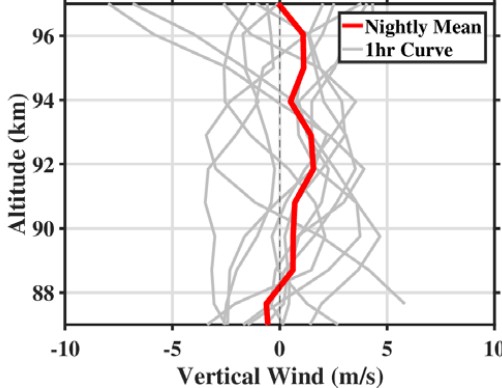

Figure 8 Vertical wind profiles of PMCW-NSL on November 24, 2021.



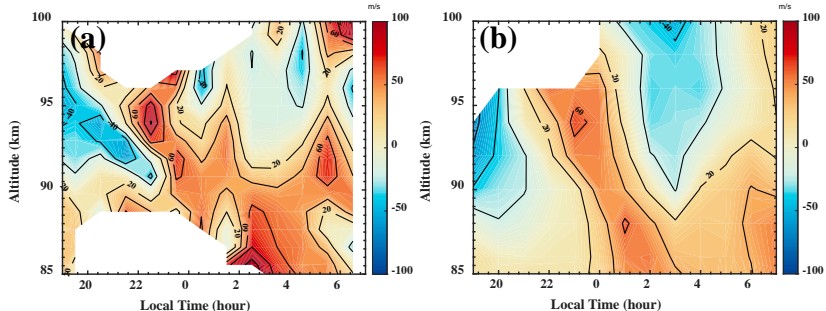

Figure 9 Zonal wind contour of (a) PMCW-NSL with a vertical resolution of 1km and temporal resolution of 1hr and (b) meteor radar with a vertical resolution of 2km and a temporal resolution of 1hr on December 1, 2021.

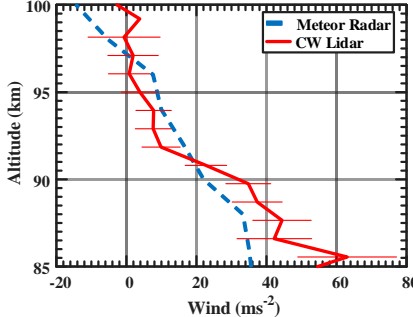

Figure 10 Comparison of nightly mean zonal wind profile between of PMCW-NSL (red solid line) and Meteor radar (blue dash line) on December 1, 2021.

Figure  compares zonal winds on December 1, 2021, observed by PMCW-NSL, with those observed by a meteor radar located at ~200km northwest of the lidar site. The uncertainty of zonal wind observed by the PMCW-NSL is less than 10 m/s at the sodium peak and ~25m /s at the edge of the sodium layer. Both observations suggest similar variations in time and structure. The downward phase progression of the semidiurnal tide is clear in both plots. The zonal wind maximum of 100 m/s observed by PMCW-NSL appears earlier than that of 80m/s by meteor radar. Figure 10 compares the nightly mean zonal wind profile of the PMCW-NSL (solid red line) and meteor radar (dashed blue line) on December 1, 2021. The zonal wind of the PMCW-NSL has a good agreement with that of the meteor radar.



**5. Summary**

We successfully developed the first PMCW-NSL system for simultaneous measurements of the mesopause region temperature and wind. The lidar system adopted a diode 1178nm seeder and a co-developed fiber Raman amplifier with a fiber-coupled AOM inside to obtain a 589nm light at 3 frequencies with almost the same power, which outputs on the same path. An EOM was introduced to

modulate the 589nm laser in M-code. Based on the innovative decoded technique and algorithm for CW lidar, both the main and the residual lights modulated by M-code are used and directed to the atmosphere in the vertical and eastward direction tilted 20° from the zenith. Two Newtonian telescopes with a diameter of 30" in corresponding pointing are used to receive the returned signals. The 3-frequency laser is emitted out sequentially triggered at 3Hz in external mode. In one period of the external trig, the special

M-code group consisted of 14 periods 127 M-code and 2 periods zero-set 127 M-code, and a short idle time is designed to acquire valid returned signals and pure background noises.

Using the PMCW-NSL system, we successfully retrieved the mesopause region temperature and wind with uncertainties of ~5K and ~10m/s at the peak of the sodium layer. The temperature comparison between PMCW-NSL observation and SABER observation shows good agreement with slight

differences. And the zonal wind results observed by PMCW-NSL also agree with those observed by a nearby meteor radar. These demonstrate that the PMCW-NSL can reliably measure the mesopause region temperature and wind. In the future, we plan to improve its accuracy and resolution by implementing a 20W 589nm CW laser in the PMCW-NSL system.

**Data availability.** The PMCW-NSL data at Hefei are available by contacting the corresponding author.

We will upload the data to the National Space Science Data Center, National Science & Technology Infrastructure of China. The wind data of the meteor radar can be applied from the National Space Science Data Center, National Science & Technology Infrastructure of China (http://www.nssdc.ac.cn/eng). The SABER temperature data are available at http://saber.gats‑inc.com/.

**Author contributions.** XF wrote the paper and provided technical guidance for lidar development. FL

developed of the lidar system and retrieved the data. LS retrieved the data. TL designed the experiment, contributed to the discussion and revision of the paper and explained of the results.

**Competing interests.** The author has declared that there are no competing interests.



**Disclaimer.** Publisher's note: Copernicus Publications remains neutral with regard to jurisdictional claims in published maps and institutional affiliations.

**Acknowledgments.** We acknowledge Wen YI provides wind data of the meteor radar, which can be applied from the National Space Science Data Center, National Science & Technology Infrastructure of China (http://www.nssdc.ac.cn/eng). We thank the SABER team for making the SABER temperature dataset available at http://saber.gats‑inc.com/. The authors would like to thank Chiao-Yao She for the helpful discussion.

**Financial support.** National Natural Science Foundation of China (42130203, 41974177, 41974175). The B-type Strategic Priority Program of the Chinese Academy of Sciences Grant No. XDB41000000. The pre-research project on Civil Aerospace Technologies No. D020105 funded by China's National Space Administration.

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
