# Peer review of "Pseudo-random modulation continuous wave narrowband sodium temperature/wind lidar"

_EGUsphere, 2022_

## Author Response (AR1)

This paper describes a new lidar and shows initial observations using the PMCW technique that has been discussed for years but has not been used operationally. The paper is well written and the results look good. I just have a few small comments and I do not need to see the paper again before publication unless other reviewers have significant concerns.

1. The writing is generally good and understandable but it could use a read over by a native English speaker to fix some prepositions, etc.
Response: Thanks for your suggestion. We have fixed some prepositions as lines 14, 20 and 40.

2. In the Introduction, the authors mention using this smaller laser for space observations. What is the effect of the large ground backscatter signal on the PMCW method. This can't easily be avoided with a large baseline like the near-field signal in a ground-based system
Response: For space observations, the large ground backscatter signal should be considered in the design of the PMCW lidar. Our preliminary idea will be to set the beam emitting in a direction tangent to the surface to avoid the large ground backscatter signal.

3. Section 2: For the EOM, what are the 0 and 1 levels? Many EOM's are 10% and 90% splitting. How does this code impurity affect the results?
Response: The 0 and 1 levels of the EOM in our lidar are about 10% and 90% respectively. The code impurity affects the results a little. But in our system, the timing is specially designed with zero-set 127 M codes after the M code sequences. Our lidar system also receives zero-code signals. Therefore, we can reduce the impact of code impurity to a certain extent.

4. Section 3: Your fiber AOM is more properly described as an AO frequency shifter, since frequency shifting is the main goal. That also avoids any confusion with your EO modulator. Similarly, figures 1 and 2 use different terms for the same device: PPLN vs SHG, you might want to pick one for clarity,
Response: Thanks for your advice. We have modified it in lines 114-115 and in Table 2 . About the PPLN, we described its function as the SHG in the text in line 118.

5. Have you sent any of the output yellow beam (before the EOM) into the Doppler Free to check for frequency offsets/broadening in the 2nd fiber amplifier?
Response: That's a good question. Our co-developer from Shanghai Frequency Calibration Co., LTD. told us that the fiber amplifier had few frequency offsets and only about a few hundred of kHz frequency broadening. But we will check them in the future.

6. For Figure 7a and 8: At 1km/1hour resolution it would be very rare to see any true vertical winds of more than 1-2 m/s, based on many observations with much higher SNR Na systems. So much of the signal in Figure 7a is likely noise. How do the

measured vertical winds compare with your PMCW error calculations? You should add error bars to Figure 8.

Response: That's right. Vertical wind generally does not exceed 1-2 m/s at 1km/1hour resolution. Since our PMCW lidar has a power of only 1.5W at 589nm, the wind measurement error due to signal statistical noise is up to about 10m/s at the sodium layer peak with vertical and temporal resolutions of 1 km and 1hr. But the error of nightly mean vertical wind decreases to 3m/s at the sodium layer peak as shown in replotted figure 8 in line 200. Here we show the nightly mean vertical wind in the reasonable range just to demonstrate the PMCW lidar system working well.

[Figure]

7. Section 5, line 260: "And the zonal wind..." -> "The zonal wind..."

Response: We have modified in line 230.

The paper presents a major upgrade and breakthrough of the USTC PMCW Na lidar. With current capability of measuring the MLT temperature and winds, this new type of Na lidar would have great potential for future spaceborne and airborne Na Doppler lidar missions. The lidar data presented in the paper have demonstrated the credibility of this new lidar technique. I will add that, in addition to the advantages listed in the paper, the so-called "chirp" issue imbedded in the pulsed lidar system (Yuan et al., 2009) is not an issue for PMCW system, because of its "pure" spectrum. My recommendation is to publish after the corrections of some minor technical issues.

Response: Thanks for your comments.

Line 75, "term the transmitted light the main light, and the reflected output light the residual light"

Response: We have modified in lines 63-64.

Line 77, "To improve the overall efficiency of the lidar,… for the residual light detection… and achieving the complete…"

Response: We have modified in lines 66-67.

Line 143, "we send the lidar laser beam at a distance…"

Response: We have modified in line 126.

Line 150, delete "to receive the backscattered returned signals"

Response: We have modified in line 135.

Line 201, "the temperature results within are…"

Response: We have modified in line 177.

Line 209, is the SABER temperature profile an instantaneous sample at the lidar station or an averaged one over an area? If it is the later, please specify the latitudinal and longitudinal range.

Response: SABER measurements temperature profile using limb remote sensing technique. It is an averaged profile over a small area. We can estimate the range from the limb geometry. At 100km tangent height, the latitudinal and longitudinal ranges are about 2º and 0.05º respectively.

Line 225, "in good agreement",

Response: We have modified in line 198.

Line 238, what is the uncertainty of the MWR wind measurement?

Response: The uncertainty of the meteor radar wind for 1hr average is less than 5m/s at 86km. We add the sentence in lines 212-213.

Line 244, "demonstrates good agreement …"

Response: We have modified in line 217.

---

## Author Response (AR2)

1) Figure 5: the two color bars need a proper title and corresponding unit: temperature (K); note that the minute "K" on top of the color bar is way too small;

Response: We have added the title "Temperature" on top of the contour plot and increased the font size for "K" on top of the color bar.

2) Figure 7: same as for Figure 5 but for vertical and zonal wind in (m/s); again the minute "m/s" on top of the color bar is unacceptably small

Response: We have added the title "Vertical wind" or "Zonal wind" on the top of the contour plot and increased the font size for 'm/s'.

3) Figure 8: error bars need to be explained either in the caption or in the text

Response: We have added one sentence description for the error bars in the text.

4) Figure 9: same as for Figures 5 and 7

Response: Done.

5) Figure 10: The unit in the x-axis title is wrong (ms^-2), should be (m/s)

Response: We have corrected the x-axis title.